# Systematic Review of Available CAR-T Cell Trials around the World

**DOI:** 10.3390/cancers14112667

**Published:** 2022-05-27

**Authors:** Luciana Rodrigues Carvalho Barros, Samuel Campanelli Freitas Couto, Daniela da Silva Santurio, Emanuelle Arantes Paixão, Fernanda Cardoso, Viviane Jennifer da Silva, Paulo Klinger, Paula do Amaral Costa Ribeiro, Felipe Augusto Rós, Théo Gremen Mimary Oliveira, Eduardo Magalhães Rego, Rodrigo Nalio Ramos, Vanderson Rocha

**Affiliations:** 1Center for Translational Research in Oncology, Instituto do Câncer do Estado de Sao Paulo, Hospital das Clínicas da Faculdade de Medicina da Universidade de Sao Paulo, Sao Paulo 01246-000, Brazil; 2Fundação Pró-Sangue–Hemocentro de Sao Paulo, Sao Paulo 05403-000, Brazil; samuel.couto@hc.fm.usp.br (S.C.F.C.); theo.gremen@hc.fm.usp.br (T.G.M.O.); vanderson.rocha@hc.fm.usp.br (V.R.); 3Laboratory of Medical Investigation in Pathogenesis and Directed Therapy in Onco-Immuno-Hematology (LIM-31), Departament of Hematology and Cell Therapy, Hospital das Clínicas da Faculdade de Medicina da Universidade de Sao Paulo, Sao Paulo 01246-000, Brazil; fernandacardoso@icb.usp.br (F.C.); viviane.jennifer@hc.fm.usp.br (V.J.d.S.); paulo.klinger@hc.fm.usp.br (P.K.); paula.acribeiro@hc.fm.usp.br (P.d.A.C.R.); felipe.ros@usp.br (F.A.R.); eduardo.rego@hc.fm.usp.br (E.M.R.); rodrigo.nalio@gmail.com (R.N.R.); 4Computational Modeling Department, National Laboratory of Scientific Computing, Petropolis 25651-075, Brazil; danielassanturio@gmail.com; 5Graduate Program, National Laboratory of Scientific Computing, Petropolis 25651-075, Brazil; earantes@lncc.br; 6Department of Immunology, Institute of Biomedical Sciences, University of Sao Paulo, Sao Paulo 05508-000, Brazil; 7Instituto D’Or de Ensino e Pesquisa, Sao Paulo 04501-000, Brazil; 8Churchill Hospital, Department of Hematology, Churchill Hospital, University of Oxford, Oxford OX3 7LE, UK

**Keywords:** hematological malignancy, clinical trial, solid malignancy, cancer, immunotherapy

## Abstract

**Simple Summary:**

CAR-T cells are genetically modified T cells that are reprogrammed to specifically eliminate cancer cells. Due to its clinical success to treat certain hematological malignancies, novel approaches to improve CAR-T cell-based therapies are being explored. This systematic review gives a worldwide overview of clinical trials evaluating new CAR-T cell therapies against different types of cancers, detailing the latest trends in CAR-T cell development.

**Abstract:**

In this systematic review, we foresee what could be the approved scenario in the next few years for CAR-T cell therapies directed against hematological and solid tumor malignancies. China and the USA are the leading regions in numbers of clinical studies involving CAR-T. Hematological antigens CD19 and BCMA are the most targeted, followed by mesothelin, GPC3, CEA, MUC1, HER2, and EGFR for solid tumors. Most CAR constructs are second-generation, although third and fourth generations are being largely explored. Moreover, the benefit of combining CAR-T treatment with immune checkpoint inhibitors and other drugs is also being assessed. Data regarding product formulation and administration, such as cell phenotype, transfection technique, and cell dosage, are scarce and could not be retrieved. Better tracking of trials’ status and results on the ClinicalTrials.gov database should aid in a more concise and general view of the ongoing clinical trials involving CAR-T cell therapy.

## 1. Introduction

The concept of harnessing the immune system to fight diseases has been described since the development of the first vaccines, dating back to the 18th century [1]. The human immune system is efficient to control both infectious and noninfectious diseases; however, cancer may escape from the immune system due to several mechanisms [2]. T cells are classically known to efficiently control tumor development through expression of tumor-specific T cell receptors that can recognize tumor antigens [3]. However, T cells with effector features can barely infiltrate and accumulate within the tumor microenvironment (TME), affecting the success of tumor control [4]. In late 1980, Dr. Zelig Eshhar’s team elaborated the idea of redirecting T cells to target antigens of choice by inserting a newly constructed receptor [5]. The first chimeric antigen receptor (CAR) was achieved by replacing the endogenous T cell receptor (TCR) variable region with the VL and VH region of an anti-2,4,6-trinitrophenyl (TNP) antibody, while maintaining the TCR extracellular constant C-region, the transmembrane, and intracellular signaling domains of T cells. Cells expressing the chimeric TCR exhibited the idiotype of Sp6 anti-TNP antibody and enabled a non-MHC-restricted response to the hapten TNP. With the evolution of biotechnology, the prototypical design of CAR genes and vectors has gone through several improvements, such as addition of costimulatory domains for better intracellular signaling, bispecific receptors that target more than one antigen, endogenous production of interleukins, immune checkpoint antagonists to enhance CAR-T cell activity, and, more recently, engineering of other immune cells such as natural killer cells, macrophages, and γδ T cells to express the CAR molecule [6,7,8].

CAR-T cell therapy is based on the ex vivo reprogramming of a patient’s own T cells with a CAR construct targeting cancer antigens. Subsequently, modified CAR-T cells are infused back into the patients, aiming to initiate an appropriate immune response and potentially eradicate tumor cells [9]. A total of six distinct CAR-T products have been approved by the American Food and Drug Administration (FDA) for clinical application since 2017. CAR-T cell products targeting the CD19 antigen have been approved for most B-cell malignancies, including relapsed or refractory (R/R) B-cell acute lymphoblastic leukemia (B-ALL), diffuse large B-cell lymphoma (DLBCL), primary mediastinal large B-cell lymphoma (PMBCL), transformed follicular lymphoma, and mantle cell lymphoma, while two anti-BCMA CAR-T cell products have been approved for R/R multiple myeloma (MM). The first clinical trials that supported the approval of CAR-T cell products started a few years ago, and we are now seeing the clinical benefits of this modality of treatment after long-term follow-up [10]. Although these products have achieved satisfactory results in clinical trials and in the real-world experience for the above-mentioned hematological malignancies, CAR-T cell therapy still lacks efficacy in other malignancies. Furthermore, the prohibitive costs of this treatment pose a real challenge to assist a wide range of patients, especially in developing countries. For this and other reasons, several clinical trials using new CAR-T cell manufacturing approaches and target-antigens are being conducted to address scientific, clinical, and cost issues to expand its use for many other oncologic diseases and to allow its use in low- and middle-income countries [11,12,13]. In this systematic review, we present data obtained from ClinicalTrials.gov (accessed on 10 March 2022) concerning CAR-T cell therapy for cancer, evidencing the worldwide geographical distribution of clinical trials, the most studied cancer types and explored combination therapies, CAR construct generations, costimulatory domains, and target-antigens.

## 2. Materials and Methods

We performed a systematic search on ClinicalTrials.gov (accessed on 10 March 2022) using the terms “CAR-T” or “chimeric antigen receptor” in October 2021 (Figure 1a). We followed the Preferred Reporting Items for Systematic Reviews and Meta-Analyses (PRISMA) guidelines. After merge and duplicate removal, 994 available clinical trials were retrieved. All 994 trials were assessed and analyzed. Only clinical trials in the oncology area were considered, excluding chronic, infectious, and autoimmune diseases. Trials that exclusively evaluated neurotoxic symptoms or long-term symptoms were also excluded because CAR-T therapy was not evaluated as primary therapy, being evaluated in another trial in multiple cases. Retrospective, follow-up, and observational studies were also excluded, as CAR-T cell therapy was not directly evaluated and was previously assessed on a former trial. Furthermore, only trials including T lymphocytes were considered, excluding other cell types such as NK cells. After exclusion of ineligible studies, we included a total of 868 studies for further analysis. Studies were classified according to study center (unicenter or multicenter), design of the trial and its funds (industry and/or public), cell source (autologous or allogeneic), CAR construct generation, costimulatory domains, number of targets, target-antigen, combination therapy, participant’s baseline disease, and primary/secondary outcomes. Appendix A contains all the information collected by the authors concerning the included and excluded clinical trials and data used in this review.

## 3. Results

### 3.1. Ongoing CAR-T Cell Therapy Clinical Trials for Oncology: Location, Type of Study, and Funding

We found that China is the country with the majority of CAR-T cell clinical trials registered on ClinicalTrials.gov (accessed on 10 March 2022), with 460 studies. The United States of America presented 286 studies, while the UK had 14, and Germany had 9. Until now, 25 countries have initiated clinical trials with CAR-T cells, but most of the research is concentrated in the northern hemisphere, while Australia, Singapore, and Malaysia have more than one registered trial (Appendix A). There are few trials being conducted in the southern hemisphere, mainly located in Latin America (Brazil and Argentina) and South Africa. These were not included as sites of active trials because they are associated with studies that are registered in another country and are not the main site of the trial. Most of the studies are on Phase I or Phase I/Phase II status, with recruiting still open (Figure 1b,c). We observed a substantial increase in the number of trials starting in 2017, and a significant decline in 2021. In addition, the great majority of CAR-T cell trials are developed in a unicenter format (Figure 1d) and using autologous production of cells (Figure 1d). Funding for studies comes mostly from the industry, followed by partnerships between the industry and federal governments, and government-only support. Moreover, studies with CAR-T cells produced by the industry (pharmaceutical companies and biotech startups) outnumber CAR-T cell products developed in academic institutions (Figure 1f). All information is available for each clinical trial at Appendix A.

### 3.2. CAR-T Clinical Trials for Hematological Malignancies

Clinical trials using CAR-T cells are more numerous for hematological malignancies in comparison to solid tumors. Among them, non-Hodgkin’s lymphoma (NHL) is still the most frequent hematological disease targeted by CAR-T cell therapy. B-ALL is the second most targeted, followed by Multiple Myeloma (MM), chronic lymphocytic leukemia (CLL), and acute myeloid leukemia (AML), while T-cell cancers and Hodgkin’s disease are still the minority (Figure 2a). CAR-T cell persistence, safety, and overall response rate are also the most common endpoints for trials in hematological malignancies (Figure 2b). We found that inhibitory small molecules that act on signaling pathways of malignant cells are the most explored combinatorial approaches of CAR-T treatment for hematological malignancies (Appendix A). There are numerous trials that combine the use of Bruton’s tyrosine kinase inhibitors (BTKi), PI3K inhibitors, proteasome inhibitors, and Ɣ-secretase inhibitors (GSI) with CAR-T treatment. Moreover, we found clinical trials that use anakinra (IL-1 inhibitor) in combination with CAR-T cell therapy to mitigate immune effector cell-associated neurotoxicity syndrome (ICANS) and cytokine release syndrome (CRS) acute toxicities.

Single-target is still the most common strategy, with approximately 100 studies using two target combinations, and few studies targeting multiple antigens (Figure 2c). Second-generation constructs are the most common, and fourth-generation CARs are more frequent for hematological cancers than third-generation constructs (Figure 2d). The 4-1BB signal is also the most common costimulatory signal (Figure 2e). In Appendix A are the ongoing clinical trials that are investigating the benefits of using CAR-T cell treatment as salvage therapy for large B-cell lymphoma (LBCL) patients with early relapse (≤12 months) following first-line treatment failure. We were also not able to retrieve information regarding the transfection method for gene delivery (viral or non-viral) for most of the assessed clinical trials.

### 3.3. CAR-T Clinical Trials for Solid Tumors

We found a total of 229 clinical studies on solid tumors. Most CAR-T cell clinical trials are recruiting patients with glioma, pancreatic, and lung cancer, but breast and prostate cancer are also under investigation (Figure 3a). Several studies did not specify the studied tumor type, using the terms “Advanced solid tumor” or “Advanced tumor”, which we grouped as “Advanced solid tumor”. It is important to note that if a study comprehends more than one tumor type, it will appear more than once (but not in the “Advanced solid tumor” category). Most studies are designed as Phase 1, with persistence, safety, and efficacy of CAR-T cells as primary endpoints, followed by overall response rate (Figure 3b). Thirty-seven trials are combining different strategies with CAR-T cell therapy (Appendix A). Administration of checkpoint blockade therapy is the most used combinatorial approach in clinical studies for solid tumors. The second most used combination includes concomitant cytokine administration. Furthermore, chemotherapy and inhibitory small molecules that are standard-of-care are being used in combination with CAR-T cells in order to improve or at least treat patients with the best possible protocol. We have also found studies using oncolytic viruses in combination with CAR-T cells for hard-to-treat solid tumors, such as pancreatic and ovarian cancers targeting the mesothelin antigen, and a variety of HER2+ tumors.

A great majority of trials in solid tumors still use single-target CAR-T cells, with only 13 studies with dual-target and 9 with more than two targets simultaneously (Figure 3c). Most of the studies did not mention CAR generation, co-stimulatory signals, or any ‘commercial brand’ that could be used to search for CAR construct, and we were not able to retrieve this information. With the available information, second-generation CAR is still the majority (Figure 3d), with 4-1BB as the most common costimulatory signal (Figure 3e). Moreover, third- and fourth-generation CARs are also common for solid malignancies. Similar to trials in hematological malignancies, missing information regarding the transfection method of CAR-T cell products for solid tumors has hindered a consensus for the most used gene delivery methodology for these malignancies.

### 3.4. Targets in Trials of Solid and Hematological Malignancies

Figure 4 shows various diseases and target-antigen combinations that are being studied in CAR-T cell trials. Each line represents a single study. On the left side is the disease and on the right side the corresponding targets. For solid tumors, the most common targets are MSLN (mesothelin), GPC3, CEA, MUC1, HER2, and EGFR. All these targets are expressed by several tumors, making them valuable for this multi-use purpose (Figure 4a). Some cancers have several studies with the same target, such as hepatocellular carcinoma with GPC3, pancreatic cancer with MSLN, and glioma with GD2. On the other hand, for hematological malignancies, targeted-antigens of the already-approved CAR-T products—CD19 and BCMA—are by far the most frequent (Figure 4b). For ALL, most studies are still applying CD19, but CD22 is emerging as a new target. AML has a great variety of targets being tested, with a preference for CD123 and CD33. For T cell leukemia/lymphoma, CD7 is the most common target, followed by CD30.

## 4. Discussion

CAR-T cell therapy is an already-implemented treatment modality in clinical practice worldwide, and it has been successfully used to treat B-cell hematological malignancies and MM. For this reason, six CAR-T cell products have been approved by the FDA—the anti-CD19 products tisagenlecleucel (tisa-cel; Kymriah^®^, Novartis, East Hanover NJ, USA), axicabtagene ciloleucel (axi-cel; Yescarta^®^, Kite A Gilead Company, Santa Monica, CA, USA), brexucabtagene autoleucel (brexu-cel; Yescarta^®^, Kite A Gilead Company), lisocabtagene maraleucel (liso-cel; JCAR017; Bristol-Myers Squibb, New York, NY, USA), and the anti-BCMA products idecabtagene vicleucel (ide-cel, Abecma; Bristol-Myers Squibb) and ciltacabtagene autoleucel (cilta-cel; Janssen Biotech, Philadelphia, PA, USA) [14,15,16,17,18,19,20,21]. However, the lack of efficient CAR-T cell treatment for hard-to-treat diseases such as AML, T-cell leukemias/lymphomas, and solid tumors represents an unmet need for patients that have failed standard-of-care treatment. With the aim to have an overview of the ongoing clinical trials of CAR-T, we performed a systematic review to evaluate the available registered clinical trials on ClinicalTrials.gov (accessed on 10 March 2022) involving CAR-T cell therapy administration in various malignancies.

From a total of 868 assessed clinical trials, we found that China is the country with the majority of ongoing clinical trials on CAR-T cells. We are aware that China has its own clinical trial repository, so the numbers are possibly underestimated. The United States is the second country with the most CAR-T cell trials, followed by the European region. It is noteworthy that a significant number of clinical trials in the US, EU, and China are funded by the industry and federal governments, which could explain the great diversity of studies in these regions when compared to low- and middle-low-income countries or regions. CAR-T therapies are currently administered at a limited number of cancer centers and are primarily delivered in the inpatient setting [22]. In addition, we noticed that pharmaceutical company and biotech start-up CAR-T cell products are largely being investigated in clinical trials, albeit academic CAR-T cell products are gaining territory and are expected to compete with commercial products. Indeed, “in-house” production of CAR-T cells can reduce the cost of centralized production due to technological improvements in production automation, and it does not require shipping and handling of the leukapheresis product, saving both time and money [23,24]. For instance, the CAR-T cell product ARI-0001 (CART19-BE-01) was the first anti-CD19 CAR that received authorization under “hospital exemption” from the Spanish drug regulatory agency to treat adult patients with R/R ALL. This was a groundbreaking approval for being the first CAR-based product to have been developed from bench to bedside in the EU. Produced in a fully academic environment but maintaining similar quality, safety, and efficacy standards of industrial manufacture, the price of ARI-001 is one-third of the commercial CAR-Ts available in Spain [25].

Autologous production of CAR-T cells is still the most used source of cells, although “universal” cells, such as HLA mismatched T-cells with TCR knockout and other immune cells, are garnering interest, and we expect to see an increase in these “off-the-shelf” products being evaluated in clinical studies in the next years [25,26]. Onco-hematologic patients, mainly those with B-cell malignancies, are the majority of enrolled participants, followed by glioma and other advanced solid tumor-bearing patients. The most commonly reported tumors, such as breast and prostate cancer, are also under investigation. Solid tumors and hematological malignancies are very distinct entities in terms of cancer profiling and TME constitution. Although the same CAR structure can be applied in both, the recognized antigen must be preferentially expressed in transformed cells or at least not expressed in vital organs [27]. Most clinical trials are Phase I and II studies and are still recruiting participants. For this reason, very few studies presented results or have published articles, which is expected to happen in the next few years. It is also noteworthy and expected that most trials are very recent, with outstanding growth after CAR-T approval for leukemia in 2017 [15]. On the other hand, we have observed a significant decrease in the number of trials in 2021, probably due to the COVID-19 pandemic [28].

We have found that the most studied primary outcomes were the evaluation of CAR-T cell administration safety profile and anti-tumoral efficacy. Secondary endpoints included evaluation of overall response to treatment, CAR-T cell persistence, and duration of response. We hypothesized that trials of CAR-T cells in solid tumors do not evaluate overall survival or progression-free survival because initial challenges on CAR-T therapy success were not achieved, and therefore any benefit could be easier to achieve than a significant impact on patient survival. Solid tumors have tumor-infiltrating lymphocytes (TILs), which have been explored as a therapy in adoptive cell transfer for the past decades with modest results. Nevertheless, the importance of immune cells in TME has been demonstrated (reviewed by [9]). The solid TME is rich in inhibitory factors, such as TGF-beta (transforming growth factor-beta), anti-inflammatory macrophages, myeloid-derived suppressor cells (MDSC), and checkpoint receptor ligands (e.g., PD-L1) [29]. All these factors could lead to CAR-T loss of function, exhaustion, and elimination, which may explain why the therapy does not work so well for solid tumors in comparison to hematologic cancers. Due to challenges such as high intra-tumoral pressure, inherent immunosuppressive TME, and ineffective CAR-T trafficking to the site of disease, these tumors represent good candidates for combination therapies [1,30].

Combination of CAR-T cell therapy with other drugs is becoming a major area of interest and we have found various ongoing trials exploring this combinatorial strategy. Immune checkpoint inhibitors, such as Ipilimumab, Nivolumab, Pembrolizumab, Tiszelimumab, and Relatlimab are the most studied drugs used in combination with CAR-T cells for solid tumors. Interestingly, with the advance of gene-editing technologies such as CRISPR-Cas9, some researchers are also evaluating the genetic modification of cells by generating single or multiple gene knockout T cells that lack expression of immune checkpoint receptors, such as PD-1, CTLA-4, LAG-3, TIGIT, and others [31,32,33]. Other combinations include the use of cytokine support, inhibitory small molecules, and oncolytic virus, to name a few. Multiple B cell malignancies are usually associated with overexpression and constitutive activation of BTK, resulting in malignant cell survival, proliferation, and migration [34,35,36,37]. BTKi such as Ibrutinib and Acalabrutinib are largely being studied in combination with CAR-T cells to improve patient treatment. Moreover, the combination of PI3K inhibitors with CAR-T therapy is based on the rationale that CAR-T cells in vivo might become rapidly exhausted upon encounter with target-antigen on tumor cells and subsequent CD28 downstream signaling. Inhibition of PI3K potentially mitigates the ill effects of exacerbated CAR-T expansion, preventing T-cell differentiation and acquisition of an exhausted phenotype [38].

CD19 and BCMA are the most targeted antigens for treating hematological malignancies, while mesothelin, CEA, HER2, and GPC3 are the main targets of interest for solid tumors. Hematologic malignancies present a lower mutational tumor burden [39] and are good candidates for therapies involving adoptive engineered T cell transfer against tumor antigens, while antigen heterogeneity and mutational load on solid tumors are probably the main challenges for CAR-T cell therapy eligibility. It is expected that the TME changes with an effective cytotoxic response, and TIL and CAR-T cell TCRs could recognize and be activated to kill other tumor cells. However, these phenomena are not enough to trigger an effective anti-tumor response in most patients. Therefore, bi-specific CAR-T cells that can recognize more than one target-antigen are being studied to circumvent this issue [40,41]. Second-generation CAR are the most explored constructs, especially for hematological malignancies, followed by fourth- and third-generation constructs. Interestingly, most fourth generation CAR-T constructs are designed to secrete monoclonal antibodies targeting checkpoint receptors, which could enhance CAR-T cell persistence and resistance to the immunosuppressive TME [42,43]. Moreover, description of the transfection methodology is absent in most of the assessed studies. Gene delivery is an essential step in the manufacture of CAR-T cells, and while most methodologies use viral-based vectors (usually retro- or lentiviruses), non-viral delivery of CAR gene has developed interest due to its production simplicity and lower cost [40,44].

Thus far, only second-generation CAR-T cells targeting two molecules, CD19 and BCMA, have been approved, with either CD28 or 4-1BB as costimulatory signals. FDA -approved CAR-T cell products are recommended for the treatment of R/R patients that have previously received multiple lines of therapy. It is possible that these products might contribute to better outcomes if they are administered as first- or second-line treatment, since each additional line of therapy is associated with lower response rates and increased comorbidities [45]. Moreover, patients with early relapse following first-line therapy have a poor prognosis, with a 50% rate of overall survival when treated with the currently available therapies [46]. Based on results of trials comparing CAR-T cell with SOC, investigators from TRANSFORM and ZUMA-7 studies support the respective CAR-T cell products as potential second-line (2L) treatment for patients with R/R DLBCL [47,48]. On the other hand, investigators of the BELINDA study state that insights from a longer follow-up and additional studies of this randomized Phase 3 trial will inform the possible use of tisagenlecleucel in the 2L R/R DLBCL setting [47].

Although this systematic review has provided a relevant take on the ongoing CAR-T cell clinical trial scenario worldwide, we found inconsistencies in the available information on the ClinicalTrials.gov repository that could represent a drawback to our study. Most registered studies lack important information that was not provided by the principal investigator—this includes CAR-T cell target-antigen, CAR construct generation, gene delivery method (e.g., viral or non-viral), number of infused CAR+ T cells, timing of administration, and other relevant information regarding the studied product. There is also a dearth of study results and participant recruitment status update, therefore hindering an adequate visualization of the number of trials that are still ongoing or have been completed, suspended, or withdrawn. For instance, we found study results that have been published in peer-reviewed journals that were not updated on ClinicalTrials.gov (accessed on 10 March 2022), thus underestimating the “completed” status of various assessed trials. Better tracking of the registered clinical trials and keeping them up to date on the ClinicalTrials.gov repository could be helpful to clinical trial researchers and patients alike.

## 5. Conclusions

This systematic review provides a general glance of CAR-T cell therapies being studied in clinical trials for hematological and solid tumors on a global scale. The USA has long been at the forefront of CAR-T cell development, but China has recently surpassed it in the number of trials for CAR-T therapy studies. There are still few studies in low- and middle-income countries, although automated and decentralized production of CAR-T cells might have a positive impact on the access to advanced cell therapies in these countries. Combination of CAR-T administration with other drugs can also improve therapy efficiency, and drugs that are regularly used in clinical settings such as ICI, BTKi, PI3K inhibitors, and oncolytic viruses are being employed for that purpose. With the recent findings that CAR-T treatment might be as efficient as SOC to treat R/R patients, CAR-T cell administration as first-line therapy does not seem so far from reality anymore. This could represent a paradigm shift in cancer patient care, and we expect that studies involving CAR-T therapy will exponentially increase in the next few years, especially with the approval of new products by regulatory agencies. Moreover, CAR-T may hold promise as a therapeutic platform not only to treat malignant diseases, but infectious and hereditary diseases alike [49,50].

## Figures and Tables

**Figure 1 cancers-14-02667-f001:**
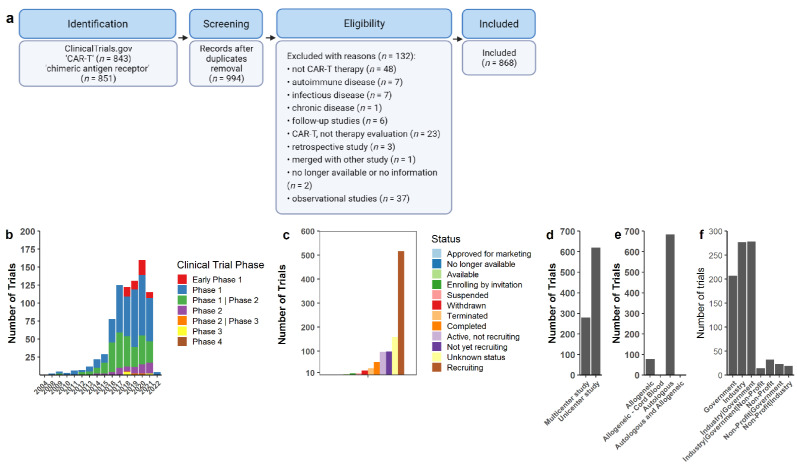
CAR-T cells on clinical trials. (**a**) Flow chart of the systematic review methodology showing the keywords, duplicate removal, and eligibility criteria applied as filters. (**b**) Number of trials by design type along the years. (**c**) Overall status of clinical trials applying CAR-T cells. (**d**) Multicenter or unicenter studies as described by each trial or observed by locations. (**e**) CAR-T cell source used in trials. The majority of trials described autologous, allogeneic cord blood (*n* = 1), and allogeneic or autologous T cells (*n* = 2). (**f**) Funding support described for clinical trials.

**Figure 2 cancers-14-02667-f002:**
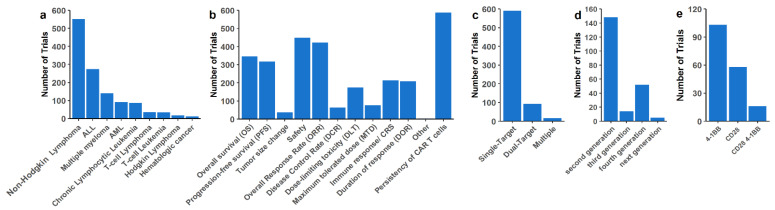
CAR-T cell strategy for hematological malignancies under clinical trial evaluation. (**a**) Hematological malignancies treated by CAR-T cells under clinical trials (non-Hodgkin’s lymphomas including DLBCL, MCL, Follicular lymphoma, mediastinal large B cell lymphoma, B cell lymphoma, CLL, and Small Lymphocytic Lymphoma). (**b**) Primary and secondary endpoints of clinical trials. All endpoints for a given trial are presented in this graph. (**c**) Number of simultaneous targets by one CAR or combination of different CAR-T cells delivered at once or in sequence for the patients. Multiple includes three or more targets at once. Several targets for multiple patients were considered as single-target therapy. (**d**) CAR generation used in clinical trials. (**e**) Number of clinical trials with CD28 and/or 4-1BB costimulatory signal on CAR constructs.

**Figure 3 cancers-14-02667-f003:**
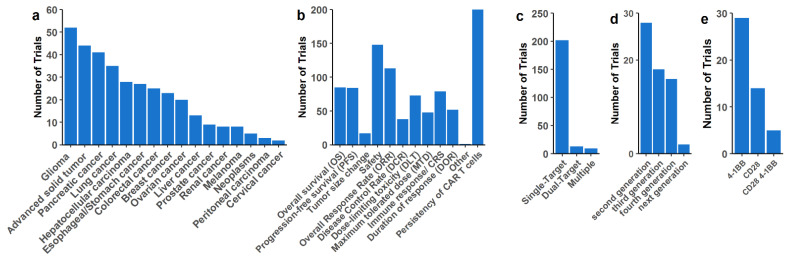
CAR-T cell strategies for solid tumors under clinical trial evaluation. (**a**) Solid tumors treated by CAR-T cells under clinical trials. The term “Advanced solid tumors” was applied when the trial description used the same terminology, and the tumor origin could not be accessed. If the trials described several tumors, all of them were marked as separated entities. (**b**) Primary and secondary endpoints of clinical trials. All endpoints for a given trial are presented in this graph. (**c**) Number of simultaneous targets by one CAR or combination of different CAR-T cells delivered at once or in sequence for the patients. ‘Multiple’ includes three or more targets at once. Several targets for multiple patients were considered as single-target therapy. (**d**) CAR generation used in clinical trials. (**e**) Number of clinical trials with CD28 and/or 4-1BB costimulatory signal on CAR constructs.

**Figure 4 cancers-14-02667-f004:**
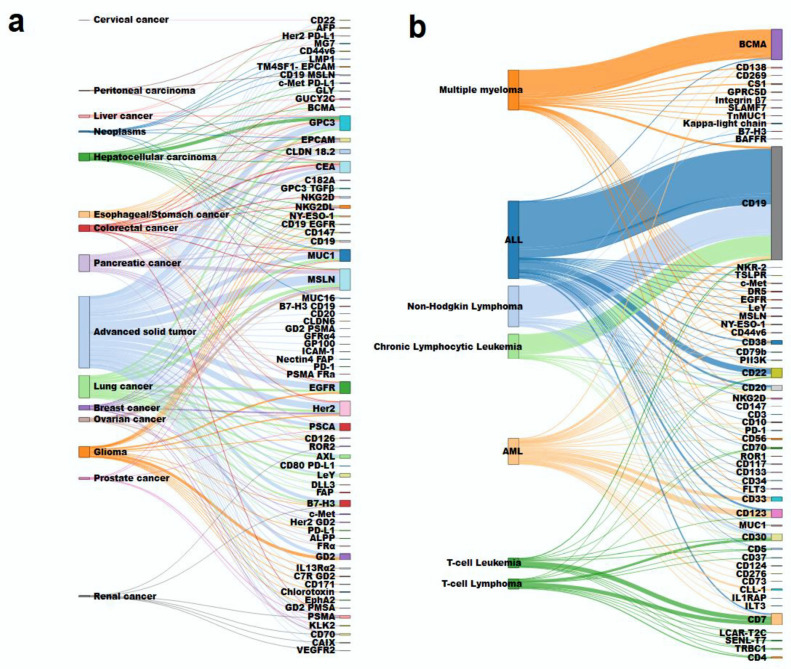
Disease and target antigen combination for CAR-T cell clinical trials. (**a**) Solid tumors are depicted on the left and the target on their right. The term “Advanced solid tumors” was attributed when the malignancy was not described in the trial. (**b**) Hematologic malignancies and the corresponding targets used in CAR-T cell trials. The line colors are according to the tumors, each line representing one study. The bar thickness is according to the number of studies of tumor or target.

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
