# Peer review of "Systematic Review of Available CAR-T Cell Trials around the World"

_cancers, 2022, doi:10.3390/cancers14112667_

Round 1

Reviewer 1 Report

This is a broad overview of available CAR-T cell trials around the world as well as emerging trends with combination immune checkpoint inhibitors.

The review does not described responses, delivery, toxicity, or challenges in great detail which I expect is beyond the scope of the intent of the review. 

Given the increasing  number of CAR-T cell trials that are likely to emerge in the next 5 years I found the review informative and useful. It is well written. 

Minor issue:

1) Figure 4 - suggest add PSMA as a target for prostate cancer and LeY for lung cancer.

Author Response

Thank you for the comments. Regarding the minor issue in Figure 4, PSMA has actually been linked to prostate cancer as well as LeY for lung cancer. Maybe the colors are not ideal, but we’ve tried our best in the color choice in this complex figure. Maybe a final image, with full resolution will be enough to check all the links. We also attached an interactive HTML version of Figures 4a and 4b.

Reviewer 2 Report

The article by Rodrigues et al. presents a summary of currently active clinical trials using CAR T-lymphocytes as a therapeutic strategy bringing an insight into the topic. Although choice algorithm of included trials is well explained, inconcistencies in results were then observed. Also, the presentation of the results could be clarified.

Several changes are suggested:

The picture 1b is redundant, it could be trasfered to supplementary data or excluded

Pictures 2c and 3c are not selfexplanatory; data would be of better use in form of a table with examples of particular "combination regimens" and explanation of presumed mechanism of benefit;  it is not clear what some terms (e.g. "co-infusion") mean. Also bridging therapy is not by definition considered as a combinatorial approach. Moreover, in design explanation, toxicity-examining trials were excluded, but corticosteroids and cytokine blokade, mostly use to prevent or treat CRS and/or ICANS are included in fig. 2c and 3c. More detailed explanation or more consistency is needed.

More detail is also needed when explaining target antigens, particularly in solid tumors. E.g. a table with most often used antigen together with particular tumors (as shown on fig. 4) could be provided together with number of trial and at least some NCT numbers of largest or most advanced trials; this would replace also fig. 2a and 3a.

At least some results (besides notoriously discussed Belinda, Transform and Zuma-7) should be provided.

Information should be more concised - in introduction only 1 anti-BCMA product is mentioned to be approved by FDA, comparing to 2 products mentioned in discussion.

Comparison of CAR T-cells therapy care with HSCT care is inadequate as the toxicity of T-cell engagers has its specificity, such a comparison should be either ommited or explained with more detail

Author Response

To answer point by point, we copied the referee`s text and highlit our answer in blue.

Referee: "The article by Rodrigues et al. presents a summary of currently active clinical trials using CAR T-lymphocytes as a therapeutic strategy bringing an insight into the topic. Although choice algorithm of included trials is well explained, inconcistencies in results were then observed. Also, the presentation of the results could be clarified."

We really appreciate the comments and modified the manuscript following the referee's suggestion hoping the results are more concise and clear.

Several changes are suggested:

The picture 1b is redundant, it could be trasfered to supplementary data or excluded

As suggested by this referee, we transferred Figure 1b to the supplementary material as Supplementary Figure 1.

Pictures 2c and 3c are not self explanatory; data would be of better use in form of a table with examples of particular "combination regimens" and explanation of presumed mechanism of benefit; it is not clear what some terms (e.g. "co-infusion") mean. Also bridging therapy is not by definition considered as a combinatorial approach. Moreover, in design explanation, toxicity-examining trials were excluded, but corticosteroids and cytokine blokade, mostly use to prevent or treat CRS and/or ICANS are included in fig. 2c and 3c. More detailed explanation or more consistency is needed.

Figures 2c and 3c are a summary of many distinct strategies of treatments used in CAR-T clinical trials, which we called treatment combinations as a general term. Following the suggestion by this referee, we remove Figures 2c and 3c and added them as Supplementary Tables 2 and 3, where individual NCT and combination strategy data can be visualized. We also change categories to avoid misunderstanding. 

More detail is also needed when explaining target antigens, particularly in solid tumors. E.g. a table with most often used antigen together with particular tumors (as shown on fig. 4) could be provided together with number of trial and at least some NCT numbers of largest or most advanced trials; this would replace also fig. 2a and 3a.

To clarify the audience, we included the data table with all gathered information as Supplementary Table 1. Each clinical trial was identified by its NCT number, including a link to the website. All the data we collected and used to display as Figures in the manuscript is also available in Supplementary Table 1. The excluded trials and exclusion reasons are also in Supplementary Table 1, as they could be of interest to the audience.

Information concerning the target antigens was also added to Supplementary Table 1, making it easier to analyze and filter trials based on target antigens or any other feature of interest. We decide to maintain Figures 2a and 3a as a form of concise presentation. 

At least some results (besides notoriously discussed Belinda, Transform and Zuma-7) should be provided.

Most of the trials deposited on clinicaltrials.gov do not have any results yet. So, we remove Table 1 and added the results in Supplementary Table 1, including the publication title and DOI whenever available, making it easier for the audience to find the relevant results.

Information should be more concised - in introduction only 1 anti-BCMA product is mentioned to be approved by FDA, comparing to 2 products mentioned in discussion.

Sorry for this misinformation, we corrected the text in the Introduction section.

Comparison of CAR T-cells therapy care with HSCT care is inadequate as the toxicity of T-cell engagers has its specificity, such a comparison should be either ommited or explained with more detail.

As CAR-T cell therapy comparison with HSCT is beyond the scope of this review, we decided to omit this sentence in the discussion section.

Reviewer 3 Report

The selection of the trials to include was well thought out.
The figures did an excellent job of summarizing the large volume of data.
The manuscript was well written and easy to read.
The manuscript gave a good summary of the current status of the CAR T cell field.

Author Response

Thank you for the kind comments, we appreciated it.

Round 2

Reviewer 2 Report

On page 3, there is remainin correcture to be done:

"study was registered in PROSPERO, registration number is xxxxxxxx."